## METHOD

# SCEPTRE improves calibration and sensitivity in single-cell CRISPR screen analysis

Timothy Barry[1], Xuran Wang[1], John A. Morris[2,3], Kathryn Roeder[1,4] and Eugene Katsevich[5*]

*Correspondence:
ekatsevi@wharton.upenn.edu
[5]Department of Statistics and Data Science, Wharton School, University of Pennsylvania, Philadelphia, PA 19104, USA
Full list of author information is available at the end of the article

## Abstract

Single-cell CRISPR screens are a promising biotechnology for mapping regulatory elements to target genes at genome-wide scale. However, technical factors like sequencing depth impact not only expression measurement but also perturbation detection, creating a confounding effect. We demonstrate on two single-cell CRISPR screens how these challenges cause calibration issues. We propose SCEPTRE: analysis of single-cell perturbation screens via conditional resampling, which infers associations between perturbations and expression by resampling the former according to a working model for perturbation detection probability in each cell. SCEPTRE demonstrates very good calibration and sensitivity on CRISPR screen data, yielding hundreds of new regulatory relationships supported by orthogonal biological evidence.

## Background

The noncoding genome plays a crucial role in human development and homeostasis: over 90% of loci implicated by GWAS in diseases lie in regions outside protein-coding exons [1]. Enhancers and silencers, segments of DNA that modulate the expression of a gene or genes in *cis*, harbor many or most of these noncoding trait loci. While millions of *cis*-regulatory elements (CREs) have been nominated through biochemical annotations, the functional role of these CREs, including the genes that they target, remain essentially unknown [2]. A central challenge over the coming decade, therefore, is to unravel the *cis*-regulatory landscape of the genome across various cell types and diseases.

Single-cell CRISPR screens (implemented by Perturb-seq [3, 4], CROP-seq [5], ECCITE-seq [6], and other protocols) are among the most promising technologies for mapping CREs to their target genes at genome-wide scale. Single-cell CRISPR screens pair CRISPR perturbations with single-cell sequencing to survey the effects of perturbations on cellular phenotypes, including the transcriptome. High multiplicity of infection (MOI) screens deliver dozens perturbations to each cell [7–9], enabling the interrogation

of hundreds or thousands of CREs in a single experiment. Single-cell screens overcome the limitations of previous technologies for mapping CREs [9]: unlike eQTLs, single-cell screens are high-resolution and can target rare variants, and unlike bulk screens, single-cell screens measure the impact of perturbations on the entire transcriptome.

Despite their promise, high-MOI single-cell CRISPR screens pose significant statistical challenges. In particular, researchers have encountered substantial difficulties in calibrating tests of association between a CRISPR perturbation and the expression of a gene. Gasperini et al. [9] found considerable inflation in their negative binomial regression-based *p*-values for negative control perturbations. Similarly, Xie et al. [8] found an excess of false-positive hits in their rank-based Virtual FACS analysis. Finally, Yang et al. [10] found that their permutation-based scMAGeCK-RRA method deems almost all gene-enhancer pairs significant in a reanalysis of the Gasperini et al. data. These works propose ad hoc fixes to improve calibration, but we argue that these adjustments are insufficient to address the issue. Miscalibrated *p*-values can adversely impact the reliability of data analysis conclusions by creating excesses of false-positive and false-negative discoveries.

In this work, we make two contributions. We (i) elucidate core statistical challenges at play in high-MOI single-cell CRISPR screens and (ii) present a novel analysis methodology to address them. We identify a key challenge that sets single-cell CRISPR screens apart from traditional differential expression experiments: the "treatment"—in this case the presence of a CRISPR perturbation in a given cell—is subject to measurement error [3, 11, 12]. In fact, underlying this measurement error are the same technical factors contributing to errors in the measurement of gene expression, including sequencing depth and batch effects. These technical factors therefore act as confounders, invalidating traditional nonparametric calibration approaches. On the other hand, parametric modeling of single-cell expression data is also fraught with unresolved difficulties.

To address these challenges, we propose SCEPTRE (analysis of Single-CEll PerTurbation screens via conditional REsampling; pronounced "scepter"). SCEPTRE is based on the conditional randomization test [13], a powerful and intuitive statistical methodology that, like parametric methods, enables simple confounder adjustment, and like nonparametric methods, is robust to expression model misspecification. We used SCEPTRE to analyze two recent, large-scale, high-MOI single-cell CRISPR screen experiments. SCEPTRE demonstrated excellent calibration and sensitivity on the data and revealed hundreds of new regulatory relationships, validated using a variety of orthogonal functional assays. In the "Discussion" section, we describe an independent work conducted in parallel to the current study in which we leveraged biobank-scale GWAS data, single-cell CRISPR screens, and SCEPTRE to dissect the *cis* and *trans* effects of noncoding variants associated with blood diseases [14]. This work highlights what we see as a primary application of SCEPTRE: dissecting regulatory mechanisms underlying GWAS associations.

## Results

### Analysis challenges

We examined two recent single-cell CRISPR screen datasets — one produced by Gasperini et al. [9] and the other by Xie et al. [8] — that exemplify several of the analysis challenges in high-MOI single-cell CRISPR screens. Gasperini et al. and Xie et al. used CRISPRi to perturb putative enhancers at high MOI in K562 cells. They sequenced

polyadenylated gRNAs alongside the whole transcriptome and assigned perturbation identities to cells by thresholding the resulting gRNA UMI counts.

Both Gasperini et al. and Xie et al. encountered substantial difficulties in calibrating tests of association between candidate enhancers and genes. Gasperini et al. computed *p*-values using a DESeq2 [15]-inspired negative binomial regression analysis implemented in Monocle2 [16], and Xie et al. computed *p*-values using Virtual FACS, a nonparametric method proposed by these authors. Gasperini et al. assessed calibration by pairing each of 50 non-targeting (or negative) control gRNAs with each protein-coding gene. These "null" *p*-values exhibited inflation, deviating substantially from the expected uniform distribution (Fig. 1a, red). To assess the calibration of Virtual FACS in a similar manner, we constructed a set of in silico negative control pairs of genes and gRNAs on the Xie et al. data (see the "Methods" section). The resulting *p*-values were likewise miscalibrated, with some pairs exhibiting strong conservative bias and others strong liberal bias (Fig. 1a, gray-green).

A core challenge in the analysis of single-cell CRISPR screens is the presence of confounders, technical factors that impact both gRNA detection probability and gene expression. The total number of gRNAs detected in a cell increases with the total number of mRNA UMIs detected in a cell ($\rho = 0.35, p < 10^{-15}$ in Gasperini et al. data;

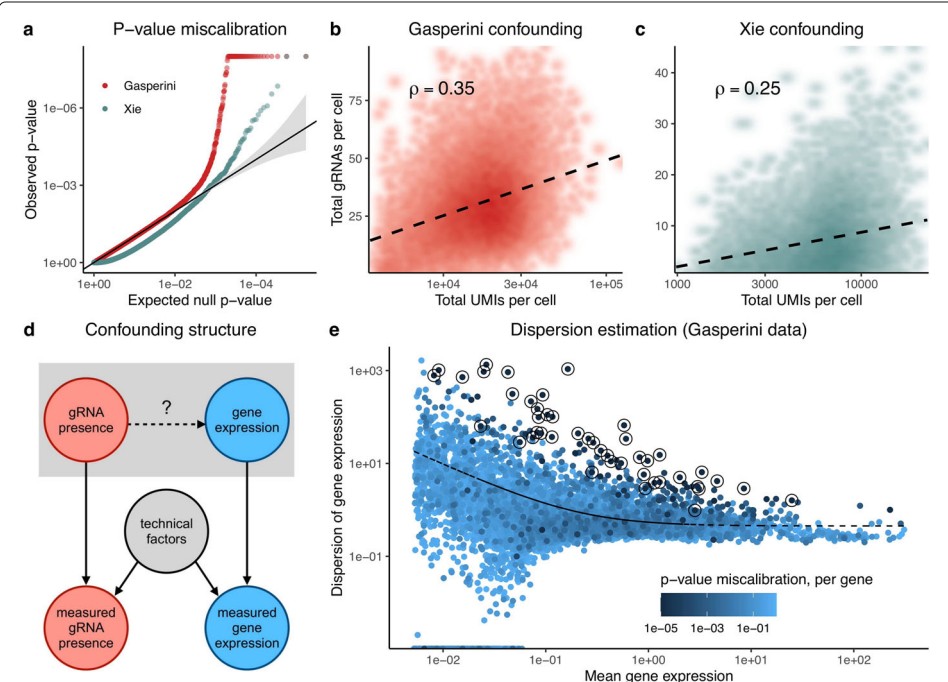

**Fig. 1** CRISPR screen analysis challenges can lead to false positives and false negatives. **a** QQ-plot of negative control *p*-values produced by Gasperini et al. (red; downsampled for visualization) and Xie et al. (gray-green). These *p*-values deviate substantially from the expected uniform distribution, indicating test miscalibration. **b–d** Technical factors, such as sequencing depth and batch, impact gRNA detection probability and observed gene expression levels in both Gasperini et al. (**b**) and Xie et al. (**c**) data. Thus, technical factors act as confounders (**d**), differentiating CRISPR screens from traditional differential expression applications. **e** Monocle2 estimates the dispersion of each gene by projecting each gene's raw dispersion estimate onto the fitted raw dispersion-mean expression curve. This estimation procedure leads to miscalibration for high-dispersion genes

$\rho = 0.25, p < 10^{-15}$ in Xie et al. data; Figs. 1b, c). Technical covariates, such as sequencing depth and batch, induce a correlation between gRNA detection probability and gene expression, even in the absence of a regulatory relationship (Fig. 1d). This confounding effect can lead to severe test miscalibration and is especially problematic for traditional nonparametric approaches, which implicitly (and incorrectly) treat cells symmetrically with respect to confounders.

Parametric regression approaches, like negative binomial regression, are the most straightforward way to adjust for confounders. However, parametric methods rely heavily on correct model specification, a challenge in single-cell analysis given the heterogeneity and complexity of the count data. We hypothesized that inaccurate estimation of the negative binomial dispersion parameter was (in part) responsible for the *p*-value inflation observed by Gasperini et al. Monocle2 estimates a raw dispersion for each gene, fits a parametric mean-dispersion relationship across genes, and finally collapses raw dispersion estimates onto this fitted line (Fig. 1e). We computed the deviation from uniformity of the negative control *p*-values for each gene using the Kolmogorov-Smirnov (KS) test, represented by the color of each point in Fig. 1e. Circled genes had significantly miscalibrated *p*-values based on a Bonferroni correction at level $\alpha = 0.05$. Genes significantly above the curve showed marked signs of *p*-value inflation, suggesting model misspecification. Analysis challenges are summarized in Table 1.

Gasperini et al. and Xie et al. incorporated ad hoc adjustments into their analyses to remedy the observed calibration issues. On closer inspection, however, these efforts were not satisfactory to ensure reliability of the results. Gasperini et al. attempted to calibrate *p*-values against the distribution negative control *p*-values instead of the more standard uniform distribution. This adjustment lead to overcorrection for some gene-enhancer pairs (false negatives) and undercorrection for others (false positives) (Fig. S1). Along similar lines Xie et al. compared their Virtual FACS *p*-values to gene-specific simulated null *p*-values to produce "significance scores" that were used to determine significance. These significance scores were challenging to interpret and could not be subjected to multiple hypothesis testing correction procedures, as they are not *p*-values.

### Improvements to the negative binomial approach

We attempted to alleviate the miscalibration within the negative binomial regression framework by following the recommendations of Hafemeister and Satija, who recently proposed a strategy for parametric modeling of single-cell RNA-seq data [17]. First, we abandoned the DESeq2-style size factors of Monocle2 and instead corrected for sequencing depth by including it as a covariate in the negative binomial regression model. Second,

**Table 1** Statistical methods employed in single-cell CRISPR screen analysis. Parametric methods are non-robust to misspecified gene expression distributions, and classical nonparametric methods cannot adjust for confounders. Conditional resampling (implemented in this work as SCEPTRE) addresses both challenges

| Method class | Example | Robust to expression model misspecification | Able to adjust for confounders |
|---|---|---|---|
| Parametric | Monocle [16] | No | Yes |
| Nonparametric | Virtual FACS [8] | Yes | No |
| Conditional resampling | SCEPTRE | Yes | Yes |

we adopted a more flexible dispersion estimation procedure: we (i) computed raw dispersion estimates for each gene, (ii) regressed the raw dispersion estimates onto the mean gene expressions via kernel regression, and (iii) projected the raw dispersion estimates onto the fitted nonparametric regression curve.

We reanalyzed the Gasperini et al. and Xie et al. negative control data using the improved negative binomial regression approach. In addition to sequencing depth, we included as covariates in the regression model the total number of expressed genes per cell and the technical factors accounted for in the original analysis (total number of gRNAs detected per cell, percentage of transcripts mapped to mitochondrial genes, and sequencing batch). Improved negative binomial regression exhibited better calibration than Monocle regression on both Gasperini et al. and Xie et al. datasets. Still, improved negative binomial regression demonstrated clear *p*-value inflation. We concluded that parametric count models likely are challenging to calibrate to high-MOI single-cell CRISPR screen data.

### SCEPTRE: analysis of single-cell perturbation screens via conditional resampling

To address the challenges identified above, we propose SCEPTRE, a methodology for single-cell CRISPR screen analysis based on the simple and powerful conditional randomization test [13] (Fig. 2). To test the association between a given gRNA and gene, we first fit the improved negative binomial statistic described above. This yields a *z*-value, which typically would be compared to a standard normal null distribution based on the parametric negative binomial model. Instead, we build a null distribution for this statistic via conditional resampling. First, we estimate the probability that the gRNA will be detected in a given cell based on the cell's technical factors, such as sequencing depth and batch. Next, we resample a large number of "null" datasets, holding gene expression and technical factors constant while redrawing gRNA assignment independently for each cell based on its fitted probability. We compute a negative binomial *z*-value for each resampled dataset, resulting in an empirical null distribution (gray histogram in Fig. 2). Finally, we compute a left-, right-, or two-tailed probability of the original *z*-value under the empirical null distribution, yielding a well-calibrated *p*-value. This *p*-value can deviate substantially from that obtained based on the standard normal (Fig. 2, Fig. S2). While we used a negative binomial regression test statistic for this work, SCEPTRE in principle is compatible with any test statistic that reasonably tracks the expression data, including, for example, statistics based on machine learning algorithms.

We leverage several computational accelerations to enable SCEPTRE to scale to large single-cell CRISPR screen datasets. First, we approximate the null histogram of the resampled test statistics using a skew-*t* distribution to obtain precise *p*-values based on a limited number of resamples (500 in the current implementation). Second, we employ statistical shortcuts that reduce the cost of each resample by a factor of about 100 (see the "Methods" section). Finally, we implement the method so that it can run in parallel on hundreds or thousands of processors on a computer cluster. (We used this approach in our independent study of noncoding blood trait GWAS loci [14].) We estimate that SCEPTRE can analyze 2.5 million gene-gRNA pairs on a dataset of 200,000 cells in a single day using 500 processors.

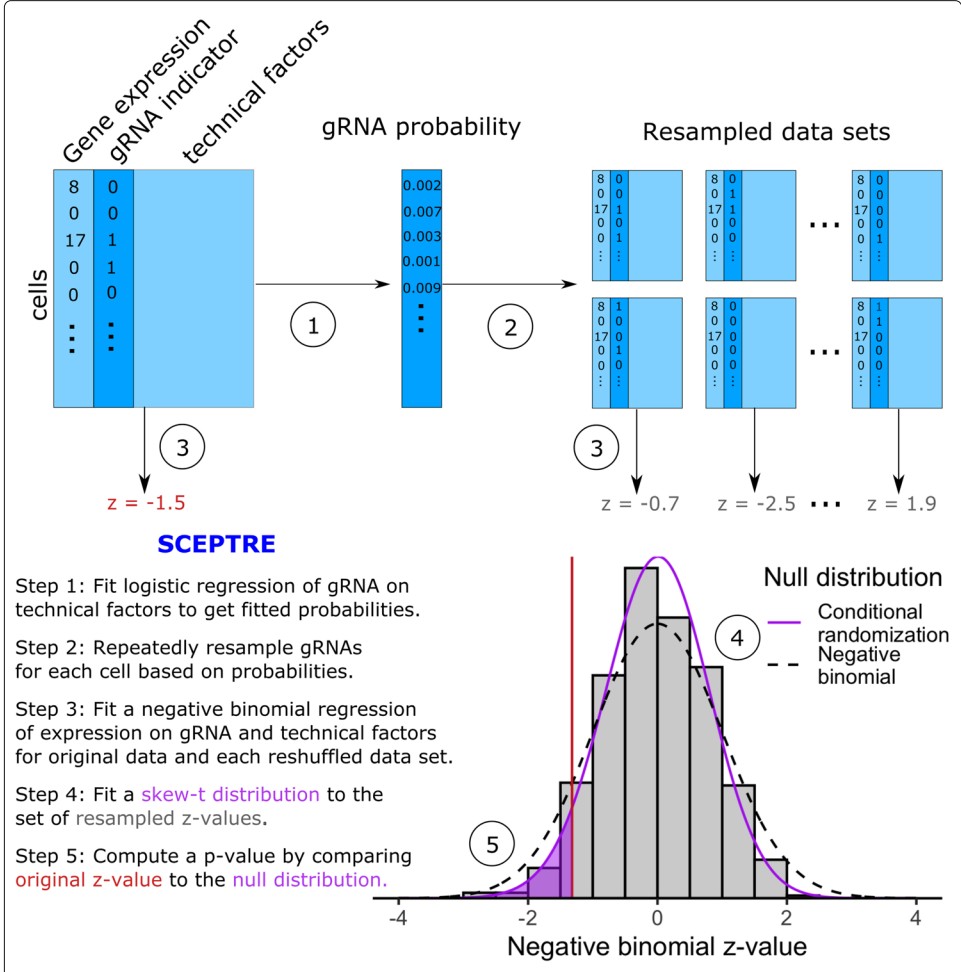

**Fig. 2** SCEPTRE: Analysis of single-cell perturbation screens via conditional resampling. A schematic and outline of the SCEPTRE methodology for one gene and one gRNA. SCEPTRE estimates the probability of gRNA detection in each cell based on its technical factors. It then builds a null distribution for the negative binomial *z*-value by independently resampling gRNA presence or absence for each cell according to these probabilities to form "negative control" datasets. A skew-*t* distribution is fit to the resulting histogram to obtain precise *p*-values based on a limited number of resamples, against which the original NB *z*-value is compared. The dashed line shows the standard normal distribution, against which the NB *z*-value typically would be compared

### SCEPTRE demonstrates good calibration and sensitivity on real and simulated data

First, we investigated the calibration of SCEPTRE in a small, proof-of-concept simulation study (Fig. 3a). We considered a class of negative binomial regression models with fixed dispersion and two technical covariates (sequencing depth and batch). We simulated expression data for a single gene in 1000 cells using four models selected from this class: the first with dispersion = 1, the second with dispersion = 0.2, the third with dispersion = 5, and the last with dispersion = 1, but with 25% zero-inflation. We also simulated negative control gRNA data using a logistic regression model with the same covariates as the gene expression model. We assessed the calibration of SCEPTRE and negative binomial regression across the four simulated datasets. To explore the impact of model misspecification on SCEPTRE and the negative binomial method (on which SCEPTRE relies), we fixed the dispersion of the negative binomial method to 1. The negative binomial

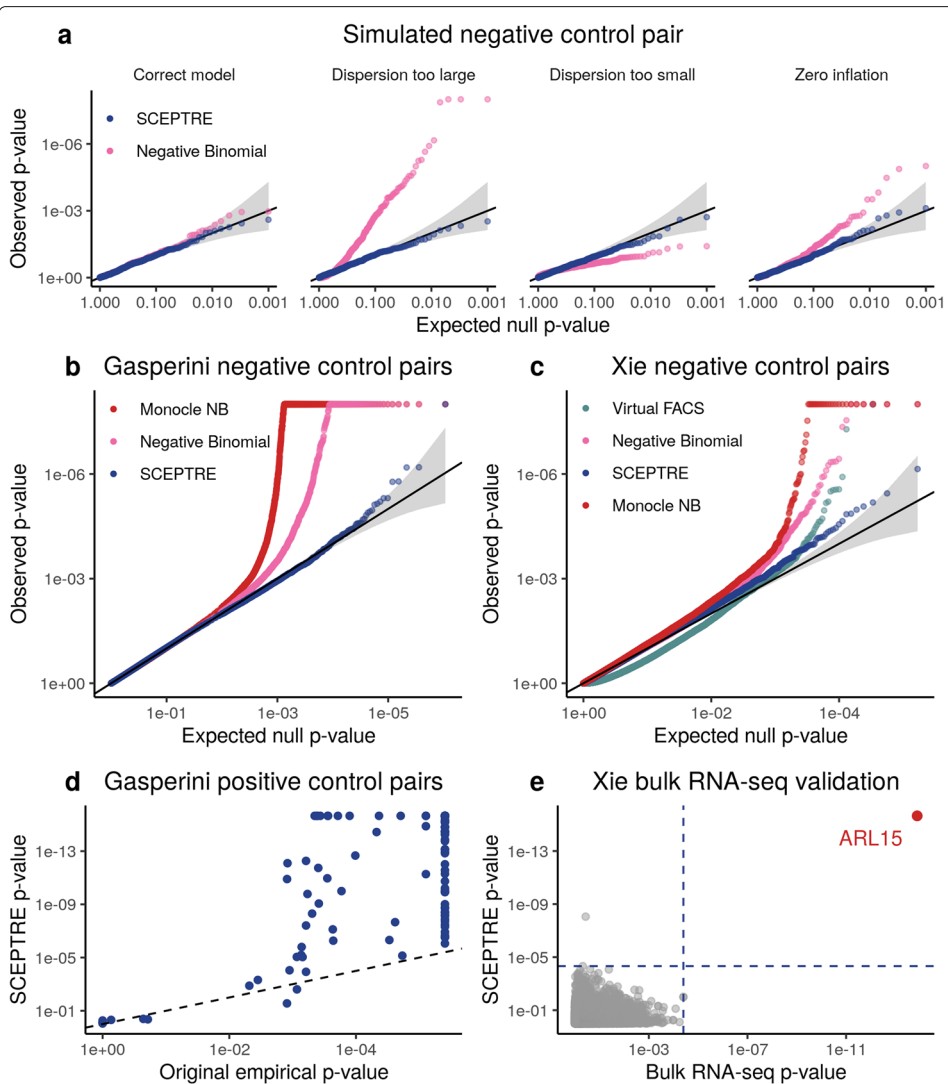

**Fig. 3** SCEPTRE demonstrates good calibration and sensitivity under known ground truth. **a** Numerical simulation comparing SCEPTRE and improved negative binomial regression on four simulated datasets. The negative binomial model was incorrectly specified on three of the four datasets. SCEPTRE maintained good calibration across all four datasets, despite model misspecification and confounder presence. **b, c** Application of SCEPTRE, improved negative binomial regression, Monocle regression, and Virtual FACS to pairs of negative control gRNAs and genes in **b** the Gasperini et al. data and **c** the Xie et al. data. Compared to the other methods, SCEPTRE showed good calibration. **d** SCEPTRE *p*-values for Gasperini et al. TSS-targeting controls were highly significant, and in general, more significant than the original empirical *p*-values. **e** Comparison of *p*-values produced by SCEPTRE for ARL15-enh to *p*-values produced by an arrayed, bulk RNA-seq CRISPR screen of ARL15-enh. The results of the two analyses coincided almost exactly, with both analyses rejecting gene *ARL15* with high confidence after a BH correction. Dotted blue lines, rejection thresholds

method worked as expected when the model was correctly specified. However, negative binomial regression broke down in all three cases of model misspecification. SCEPTRE demonstrated good calibration in all settings.

Next, to assess the calibration of SCEPTRE on real data, we applied SCEPTRE to test the association between negative control gRNAs and genes in the Gasperini et al. data (Fig. 3b) and Xie et al. data (Fig. 3c). We compared SCEPTRE to Monocle regression and the improved negative binomial method. For the Xie et al. data, we also compared to

Virtual FACS, the method originally applied to the data. SCEPTRE showed good calibration on both datasets; by contrast, Monocle regression and improved negative binomial regression demonstrated signs of severe $p$-value inflation, while Virtual FACS exhibited a bimodal $p$-value distribution peaked at 0 and 1.

SCEPTRE demonstrated modestly better calibration on the Gasperini et al. data than on the Xie et al. data. This likely is because the Gasperini et al. negative control pairs — which consisted of real, non-targeting gRNAs — were higher-quality than the Xie et al. negative control pairs — which were constructed in silico using enhancer-targeting gRNAs (see the "Methods" section). We reasoned that the Xie et al. negative controls carried mild regulatory signal, resulting in slight inflation of the SCEPTRE $p$-values on these data relative to the Gasperini et al. data.

To assess the sensitivity of SCEPTRE, we applied SCEPTRE to test the 381 positive control pairs of genes and TSS-targeting gRNAs assayed by Gasperini et al. (Fig. 3d). Allowing for the fact that the empirical correction employed by Gasperini et al. limited the accuracy of $p$-values to about $10^{-6}$, the SCEPTRE $p$-values for the positive controls were highly significant, and in particular, almost always more significant than the original empirical $p$-values, indicating greater sensitivity. Finally, we assessed the sensitivity of SCEPTRE on the Xie et al. data. Xie et al. conducted an arrayed CRISPR screen with bulk RNA-seq readout of ARL15-enh, a putative enhancer of gene *ARL15*. Both SCEPTRE and the bulk RNA-seq differential expression analysis rejected *ARL15* at an FDR of 0.1 after a Benjamini-Hochberg (BH) correction, increasing our confidence in the calibration and sensitivity of SCEPTRE (Fig. 3e).

### Analysis of candidate *cis*-regulatory pairs

We applied SCEPTRE to test all candidate *cis*-regulatory pairs in the Gasperini et al. ($n = 84,595$) and Xie et al. ($n = 5,209$) data. A given gene and gRNA were considered a "candidate pair" if the gRNA targeted a site within one Mb the gene's TSS. SCEPTRE discovered 563 and 139 gene-enhancer links at an FDR of 0.1 on the Gasperini et al. and Xie et al. data, respectively. We used several orthogonal assays to quantify the enrichment of SCEPTRE's discovery set for regulatory biological signals, and we compared the SCEPTRE results to those of other methods.

SCEPTRE's discovery set on the Gasperini et al. data was highly biologically plausible, and in particular, more enriched for biological signals of regulation than the original discovery set. Gasperini et al. discovered 470 gene-gRNA pairs at a reported FDR of 0.1. The SCEPTRE $p$-values and original empirical $p$-values diverged substantially: of the 670 gene-enhancer pairs discovered by either method, SCEPTRE and the original method agreed on only 363, or 54% (Fig. 4a). Gene-enhancer pairs discovered by SCEPTRE were physically closer (mean distance = 65 kb) to each other than those discovered by the original method (mean distance = 81 kb; Fig. 4b). Furthermore, SCEPTRE's gene-enhancer pairs fell within the same topologically associating domain (TAD) at a higher frequency (74%) than the original pairs (71%). Pairs within the same TAD showed similar levels of HI-C interaction frequency across methods, despite the fact that SCEPTRE discovered 85 more same-TAD pairs (Fig. 4c). Finally, enhancers discovered by SCEPTRE showed improved enrichment across all eight cell-type relevant ChIP-seq targets reported by Gasperini et al. (Fig. 4d, Fig. S5a).

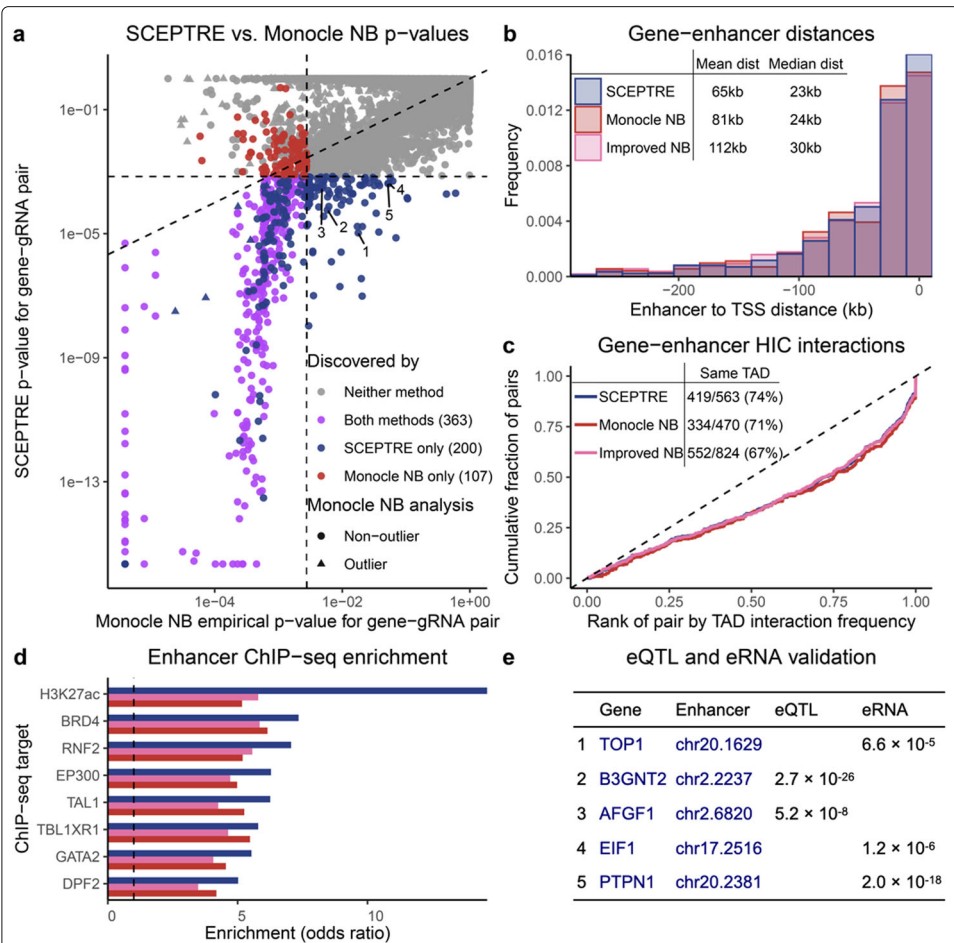

**Fig. 4** Application of SCEPTRE to Gasperini et al. data yields biologically plausible gene-enhancer links. **a** Comparison of the original empirical *p*-values to those obtained from SCEPTRE. The two analysis methods differed substantially, with 200 gene-enhancer links discovered only by SCEPTRE and 107 discovered only by the original analysis. Annotations correspond to pairs in panel (**e**). **b** Distribution of distances from TSS to upstream paired enhancers. Compared to Monocle NB (original) and improved NB analyses, SCEPTRE paired genes with nearer enhancers on average. **c** For those gene-enhancer pairs falling in the same TAD, the cumulative distribution of the fractional rank of the HI-C interaction frequency compared to other distance-matched loci pairs within the same TAD. SCEPTRE showed similar enrichment despite finding 25% more within-TAD pairs compared to the original analysis. Inset table shows gene-enhancer pairs falling in the same TAD. SCEPTRE found 93 more total pairs compared to the original analysis, and a higher percentage of pairs fell within the same TAD. **d** Enrichment of ChIP-seq signal from seven cell-type relevant transcription factors and one histone mark (H3K27ac) among paired enhancers. SCEPTRE showed stronger enrichment across all ChIP-seq targets. **e** Five gene-enhancer pairs discovered by SCEPTRE but not the original analysis, each supported by a whole blood GTEx eQTL or FANTOM enhancer RNA correlation *p*-value

When we compared discoveries unique to SCEPTRE ($n = 200$) against those unique to the original method ($n = 107$), the disparities became more extreme (Fig. S3). For example, only 57% of pairs unique to the original method fell within the same TAD, compared to 73% unique to SCEPTRE. We concluded that many pairs in the Gasperini et al. discovery set likely were false positives. Finally, when we compared SCEPTRE to the improved negative binomial method ($n = 824$ discoveries), we observed even greater differences in discovery set quality in favor of SCEPTRE (Figs. 4b–d).

We highlight several especially interesting gene-enhancer pairs discovered by SCEPTRE. Five discoveries (Fig. 4a, labels 1–5; Fig. 4e) were nominated as probable

gene-enhancer links by eQTL [18] and eRNA [19] *p*-values in relevant tissue types. The SCEPTRE *p*-values for these pairs were 1–2 orders of magnitude smaller than the original empirical *p*-values, hinting at SCEPTRE's greater sensitivity. Additionally, six pairs (Fig. 4a, blue triangles) were discovered by SCEPTRE but discarded as outliers by the original analysis, underscoring SCEPTRE's ability to handle genes with arbitrary expression distributions.

We repeated the same orthogonal analyses for the SCEPTRE discoveries on the Xie et al. data, comparing SCEPTRE's results to those of Xie et al. Xie et al.'s analysis method, Virtual FACS, outputted "significance scores" rather than *p*-values (see the "Analysis challenges" section). Because significance scores cannot be subjected to multiple hypothesis testing correction procedures (like BH), we compared the top 139 Virtual FACS pairs (ranked by significance score) against the set of 139 (FDR = 0.1) SCEPTRE discoveries (Fig. 5a; see the "Methods" section). Of the 195 pairs in either set, SCEPTRE and Virtual FACS agreed on only 83, or 43%. The SCEPTRE discoveries were more biologically plausible: compared to the Virtual FACS pairs, the SCEPTRE pairs were (i) physically closer (Fig. 5b), (ii) more likely to fall within the same TAD (Fig. 5c), (iii) more likely to interact when in the same TAD (Fig. 5c), and (iv) more enriched for all eight cell-type relevant ChIP-seq targets (Fig. 5d, Fig. S5b). When we examined the symmetric difference of the discovery sets, these differences became more pronounced (Fig. S4).

We additionally compared SCEPTRE to Monocle regression ($n = 180$ discoveries) and improved negative binomial regression ($n = 156$ discoveries) on the Xie et al. data. SCEPTRE uniformly dominated Monocle: SCEPTRE pairs were physically closer to one another (median distance = 44kb versus 110kb; Fig. 5b); SCEPTRE pairs interacted more frequently and were more likely to fall within the same TAD (68% versus 61%; Fig. 5c); and SCEPTRE pairs were more enriched for seven of eight cell type-relevant ChIP-seq targets (one target, DP22, was a tie; Fig. 5d). Improved negative binomial regression was more competitive than Monocle across metrics (Fig. 5b–d). However, as noted earlier, improved negative binomial regression exhibited severe miscalibration on the negative control pairs (Fig. 3d), rendering its discovery set less reliable than that of SCEPTRE.

### Gene expression level and sensitivity

To better understand when SCEPTRE works best, we investigated the impact of gene expression level on the sensitivity of SCEPTRE. We binned candidate gene-enhancer pairs into non-overlapping categories based on mean expression level of the gene. On both the Gasperini et al. and Xie et al. data, we found that candidate pairs containing a highly expressed gene were more likely to be rejected than candidate pairs containing a lowly-expressed gene (Tables S1, S2), indicating SCEPTRE's greater sensitivity for highly expressed genes. We observed similar trends for the other methods (not shown), consistent with the intuition that highly-expressed genes carry more information.

### Discussion

In this work we illustrated a variety of statistical challenges arising in the analysis of high-MOI single-cell CRISPR screens, leaving existing methods (based on parametric expression models, permutations, or negative control data) vulnerable to miscalibration. To address these challenges, we developed SCEPTRE, a resampling method based on modeling the probability a given gRNA will be detected in a given cell, based on that

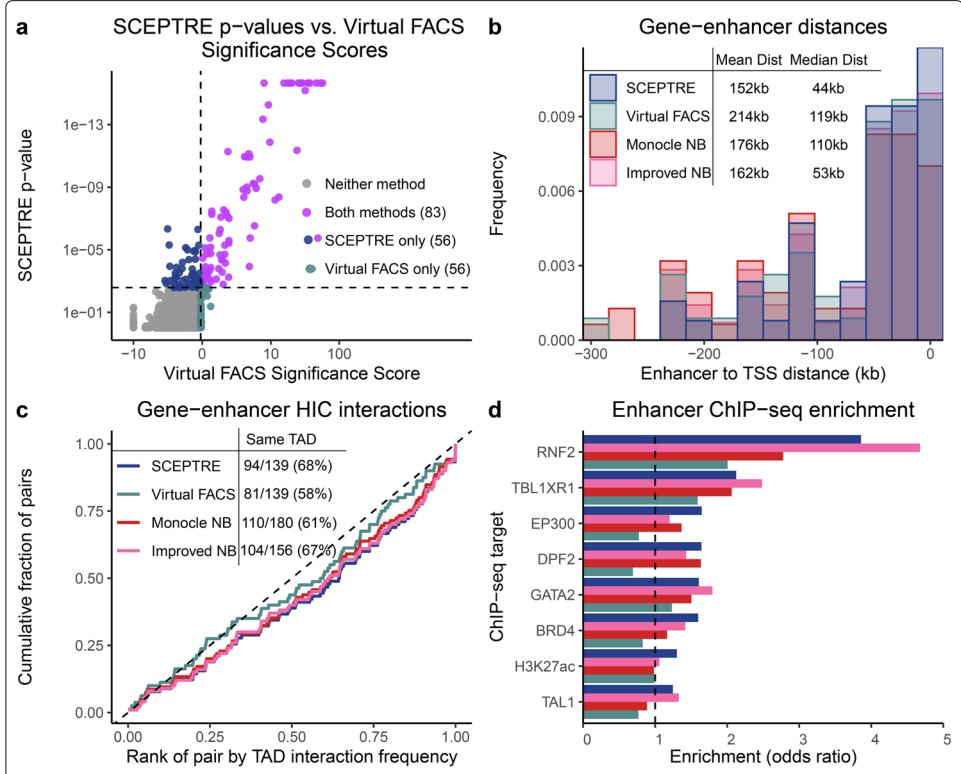

**Fig. 5** SCEPTRE discovers biologically plausible gene-enhancer links on Xie et al. data. **a** Comparison of SCEPTRE *p*-values to Virtual FACS significance scores. Significant SCEPTRE *p*-values (*n* = 139) are colored in blue and purple, and the top 139 Virtual FACS pairs, as ranked by significance score, are colored in gray-green and purple. The two sets diverged substantially, with only 43% of pairs in shared across sets. **b–d** These panels are similar to the corresponding panels in Fig. 4. SCEPTRE pairs showed strong enrichment for biological signals associated with enhancer activity on **b** physical distance, **c** HIC interaction, and **d** ChIP-seq metrics relative to other methods

cell's technical factors. We found that SCEPTRE exhibited very good calibration despite the presence of confounding technical factors and misspecification of single-cell gene expression models. We implemented computational accelerations to bring the cost of the resampling-based methodology down to well within an order of magnitude of the traditional negative binomial parametric approach, making it quite feasible to apply for large-scale data. We used SCEPTRE to reanalyze the Gasperini et al. and Xie et al. data. While our analysis replicated many of their findings, it also clarified other relationships, removing a large set (> 20% for Gasperini) of pairs that exhibited a weak relationship and adding an even larger set (> 40% for Gasperini) of new, biologically plausible gene-enhancer relationships. These links were supported by orthogonal evidence from eQTL, enhancer RNA co-expression, ChIP-seq, and HI-C data.

As an example application of SCEPTRE, we highlight *STING-seq*, a platform that we developed in parallel to the current work in an independent study [14]. STING-seq leverages biobank-scale GWAS data and single-cell CRISPR screens to map noncoding, disease-associated variants at scale. First, we used statistical fine-mapping to identify a set of 88 putatively causal variants from 56 loci associated with quantitative blood traits. We perturbed the selected variants at high MOI in K562 cells using an improved CRISPRi platform and sequenced gRNAs and transcriptomes in individual cells using ECCITE-seq

[6], a protocol that enables the profiling of multiple modalities and the direct capture of gRNAs.

We then applied SCEPTRE to quantify associations between perturbations and changes in gene expression in *cis* (within 500 kb) and *trans*. SCEPTRE confidently mapped 37 noncoding variants to their *cis* target genes, in some cases identifying a causal variant among a set of candidate variants in strong LD. Nine variants were found to regulate a gene other than the closest gene, and four variants were found to regulate multiple genes, an apparent example of pleiotropy. Several perturbations lead to widespread changes in gene expression, illuminating *trans*-effects networks. For example, two variants that were found to regulate the transcription factor *GFI1B* in *cis* altered the expression of hundreds of genes in *trans* upon perturbation; these differentially expressed genes were strongly enriched for GFI1B binding sites and blood disease GWAS hits. We concluded on the basis of this study SCEPTRE can power the systematic dissection regulatory networks underlying GWAS associations.

Despite these exciting results, key challenges remain in the analysis of single-cell CRISPR screens. Currently, SCEPTRE does not estimate the effect sizes of perturbations, disentangle interactions among perturbed regulatory elements [20, 21], or leverage information from off-targeting gRNAs to improve power. Such extensions could be implemented by harnessing more sophisticated, multivariate models of gRNA detection or applying methods for estimating variable importance in the presence of possibly misspecified models [22]. The statistical challenges that we identified in this study — specifying an accurate expression model and accounting for technical factors — and the solutions that we proposed — conditional resampling and massively parallel association testing — will help guide the development of future versions of SCEPTRE.

## Conclusions

Single-cell CRISPR screens will play a key role in unraveling the regulatory architecture of the noncoding genome [23]. Technological improvements and methodological innovations will increase the scope, scale, and variety of theses screens over the coming years. For example, screens of candidate CREs could be extended to different, disease-relevant cell types and tissues (although this remains a challenge); new combinatorial indexing strategies, such as scifi-RNA-seq, could enable the scaling-up of such screens to millions of cells [24]; different CRISPR technologies, such as CRISPRa, could enable the activation, rather than repression, of candidate CREs, yielding new insights; and information-rich, multimodal single-cell readouts could strengthen conclusions drawn about regulatory relationships [25]. SCEPTRE is a flexible, robust, and efficient method: it has now successfully been applied to three single-cell CRISPR screen datasets, across two technologies (CROP-seq and ECCITE-seq), to map regulatory relationships both in *cis* and in *trans*. We expect SCEPTRE to facilitate the analysis of future single-cell screens of the noncoding genome, advancing understading of CREs and enabling the detailed interpretation of GWAS results.

## Methods

### Gasperini et al. and Xie et al. data

Gasperini et al. used CROP-seq [5, 11] to transduce a library of CRISPR guide RNAs (gRNAs) into a population of 207,324 K562 cells expressing the Cas9-KRAB repressive

complex at a high multiplicity of infection. Each cell received an average of 28 perturbations. The gRNA library targeted 5779 candidate enhancers, 50 negative controls, and 381 TSS-targeting positive controls. Xie et al. used Mosaic-seq [7, 8] to perturb at a high multiplicity of infection 518 putative enhancers in a population of 106,670 Cas9-KRAB-expressing K562 cells. Each putative enhancer was perturbed in an average of 1276 cells.

### *Cis* and in silico negative control pairs for Xie et al. data

We generated the set of candidate *cis* gene-enhancer relationships on the Xie et al. data by pairing each protein-coding gene with each gRNA targeting a site within 1 Mb of the TSS of the gene. This procedure resulted in 3553 candidate *cis* gene-enhancer links that we tested using SCEPTRE and Virtual FACS.

To generate the set of in silico negative control pairs for calibration assessment, we (i) identified gRNAs that targeted sites far (> 1 Mb) from the TSSs of known transcription factor genes and (ii) paired these gRNAs with genes located on other chromosomes. We excluded all pairs containing genes known to be transcription factors, and we downsampled the pairs so that each gRNA was matched to 500 genes. The final in silico negative control set consisted of 84,500 pairs, the elements of which were not expected to exhibit a regulatory relationship.

### Conditional randomization test

Consider a particular gene-gRNA pair. For each cell $i = 1, \ldots, n$, let $X_i \in \{0, 1\}$ indicate whether the gRNA was present in the cell, let $Y_i \in \{0, 1, 2, \ldots\}$ be the gene expression in the cell, defined as the number of unique molecular identifiers (UMIs) from this gene, and let $Z_i \in \mathbb{R}^d$ be a list of cell-level technical factors. Letting $(X, Y, Z) = \{(X_i, Y_i, Z_i)\}_{i=1}^n$, consider any test statistic $T(X, Y, Z)$ measuring the effect of the gRNA on the expression of the gene. The conditional randomization test [13] is based on resampling the gRNA indicators independently for each cell. Letting $\pi_i = \mathbb{P}[X_i = 1 | Z_i]$, define random variables

$$\widetilde{X}_i \overset{\text{ind}}{\sim} \text{Ber}(\pi_i). \tag{1}$$

Then, the CRT $p$-value is given by

$$p_{\text{CRT}} = \mathbb{P}[T(\widetilde{X}, Y, Z) \geq T(X, Y, Z) \mid X, Y, Z]. \tag{2}$$

This translates to repeatedly sampling $\widetilde{X}$ from the distribution (1), recomputing the test statistic with $X$ replaced by $\widetilde{X}$, and defining the $p$-value as the probability the resampled test statistic exceeds the original. Under the null hypothesis that the gRNA perturbation does not impact the cell (adjusting for technical factors), i.e., $Y \perp\!\!\!\perp X \mid Z$, we obtain a valid $p$-value (2), *regardless of the expression distribution $Y|X, Z$ and regardless of the test statistic $T$.* We choose as a test statistic $T$ the $z$-score of $X_i$ obtained from a negative binomial regression of $Y_i$ on $X_i$ and $Z_i$:

$$Y_i \overset{\text{ind}}{\sim} \text{NegBin}(\mu_i, \alpha); \quad \log(\mu_i) = \beta_0 + X_i \beta + Z_i^T \gamma, \tag{3}$$

where $\alpha$ is the dispersion. Following Hafemeister and Satija [17], we estimate $\alpha$ by pooling dispersion information across genes, and we include sequencing depth as an entry in the vector of technical factors $Z_i$ (see the "Improvements to the negative binomial approach" section).

### Accelerations to the conditional randomization test

We implemented computational accelerations to the conditional randomization test. First, we employed the recently proposed [26] *distillation* technique to accelerate the recomputation of the negative binomial regression for each resample. The idea is to use a slightly modified test statistic, consisting of two steps:

1  Fit $(\widehat{\beta}_0, \widehat{\gamma})$ from the negative binomial regression (3) except without the gRNA term:

$$Y_i \overset{\text{ind}}{\sim} \text{NegBin}(\mu_i, \alpha); \quad \log(\mu_i) = \beta_0 + Z_i^T \gamma. \tag{4}$$

2  Fit $\widehat{\beta}$ from a negative binomial regression with the estimated contributions of $Z_i$ from step 1 as offsets:

$$Y_i \overset{\text{ind}}{\sim} \text{NegBin}(\mu_i, \alpha); \quad \log(\mu_i) = X_i\beta + \widehat{\beta}_0 + Z_i^T \widehat{\gamma}. \tag{5}$$

Conditional randomization testing with this two-step test statistic, which is nearly identical to the full negative binomial regression (3), is much faster. Indeed, since the first step is not a function of $X_i$, it remains the same for each resampled triple $(\widetilde{X}, Y, Z)$. Therefore, only the second step must be recomputed with each resample, and this step is faster because it involves only a univariate regression.

Next, we accelerated the second step above using the sparsity of the binary vector $(X_1, \ldots, X_n)$ (or a resample of it). To do so, we wrote the log-likelihood of the reduced negative binomial regression (5) as follows, denoting by $\ell(Y_i, \log(\mu_i))$ the negative binomial log-likelihood:

$$\sum_{i=1}^{n} \ell(Y_i, X_i\beta + \widehat{\beta}_0 + Z_i^T \widehat{\gamma}) = \sum_{i:X_i=0} \ell(Y_i, \widehat{\beta}_0 + Z_i^T \widehat{\gamma}) + \sum_{i:X_i=1} \ell(Y_i, \beta + \widehat{\beta}_0 + Z_i^T \widehat{\gamma})$$
$$= C + \sum_{i:X_i=1} \ell(Y_i, \beta + \widehat{\beta}_0 + Z_i^T \widehat{\gamma}).$$

This simple calculation shows that, up to a constant that does not depend on $\beta$, the negative binomial log-likelihood corresponding to the model (5) is the same as that corresponding to the model with only intercept and offset term for those cells with a gRNA:

$$Y_i \overset{\text{ind}}{\sim} \text{NegBin}(\mu_i, \alpha); \quad \log(\mu_i) = \beta + \widehat{\beta}_0 + Z_i^T \widehat{\gamma}, \quad \text{for } i \text{ such that } X_i = 1. \tag{6}$$

The above negative binomial regression is therefore equivalent to Eq. 5, but much faster to compute, because it involves much fewer cells. For example, in the Gasperini et al. data, each gRNA is observed in only about 1000 of the 200,000 total cells.

### SCEPTRE methodology

In practice, we must estimate the gRNA probabilities $\pi_i$ as well as the $p$-value $p_{\text{CRT}}$. This is because usually we do not know the distribution $X|Z$ and cannot compute the conditional probability in Eq. 2 exactly. We propose to estimate $\pi_i$ via logistic regression of $X$ on $Z$, and to estimate $p_{\text{CRT}}$ by resampling $\widetilde{X}$ a large number of times and then fitting a skew-$t$ distribution to the resampling null distribution $T(\widetilde{X}, Y, Z)|X, Y, Z$. We outline SCEPTRE below:

1  Fit technical factor effects $(\widehat{\beta}_0, \widehat{\gamma})$ on gene expression using the negative binomial regression (4).
2  Extract a $z$-score $z(X, Y, Z)$ from the reduced negative binomial regression (6).

3　Assume that

$$X_i \overset{\text{ind}}{\sim} \text{Ber}(\pi_i); \quad \log\left(\frac{\pi_i}{1-\pi_i}\right) = \tau_0 + Z_i^T \tau \tag{7}$$

for $\tau_0 \in \mathbb{R}$ and $\tau \in \mathbb{R}^d$, and fit $(\widehat{\tau}_0, \widehat{\tau})$ via logistic regression of $X$ on $Z$. Then, extract the fitted probabilities $\widehat{\pi}_i = (1 + \exp(-(\widehat{\tau}_0 + Z_i^T\widehat{\tau})))^{-1}$.

4　For $b = 1, \ldots, B$,

- Resample the gRNA assignments based on the probabilities $\widehat{\pi}_i$ to obtain $\widetilde{X}^b$ (1).
- Extract a $z$-score $z(\widetilde{X}^b, Y, Z)$ from the reduced negative binomial regression (6).

5　Fit a skew-$t$ distribution $\widehat{F}_{\text{null}}$ to the resampled $z$-scores $\{z(\widetilde{X}^b, Y, Z)\}_{b=1}^B$.

6　Return the $p$-value $\widehat{p}_{\text{SCEPTRE}} = \mathbb{P}[\widehat{F}_{\text{null}} \leq z(X, Y, Z)]$.

In our data analysis, we used $B = 500$ resamples.

### Numerical simulation to assess calibration

We simulated one gene $Y_i$, one gRNA $X_i$, and two confounders $Z_{i1}, Z_{i2}$ in $n = 1000$ cells. We generated the confounders $Z_{i1}$ and $Z_{i2}$ by sampling with replacement the batch IDs and log-transformed sequencing depths of the cells in the Gasperini dataset. The batch ID confounder $Z_{i1}$ was a binary variable, as the Gasperni data included two batches. Next, we drew the gRNA indicators $X_i$ i.i.d. from the logistic regression model (7), with $\tau_0 = -7, \tau_1 = -2$, and $\tau_2 = 0.5$. We selected these parameters to make the probability of gRNA occurrence about 0.04 across cells. Finally, we drew the gene expression $Y_i$ from the following zero-inflated negative binomial model:

$$Y_i \sim \lambda\delta_0 + (1 - \lambda)\text{NegBin}(\mu_i, \alpha), \log(\mu_i) = \beta_0 + Z_i^T\beta.$$

Note that gRNA presence or absence does not impact gene expression in this model. We set $\beta_0 = -2.5, \beta_1 = -2, \beta_2 = 0.5$ to make the average gene expression about 4 across cells. We generated the four datasets shown in Fig. 3a by setting the dispersion parameter $\alpha$ and the zero inflation rate parameter $\lambda$ equal to the following values:

$$(\lambda_1, \alpha_1) = (0, 1); \ (\lambda_2, \alpha_2) = (0, 5); \ (\lambda_3, \alpha_3) = (0, 0.2); \ (\lambda_4, \alpha_4) = (0.25, 1).$$

For the first, the negative binomial model is correctly specified. For the second and third, the dispersion estimate of 1 is too small and too large, respectively. The last setting exhibits zero inflation. We applied SCEPTRE and negative binomial regression to the four problem settings, each with $n_{sim} = 500$ repetitions. The negative binomial method, and in turn SCEPTRE, was based on the $z$ statistic from the Hafemeister-inspired negative binomial model (3) with $\alpha = 1$. We used $B = 500$ resamples for SCEPTRE, the default choice.

### scMAGeCK

scMAGeCK-LR [10] is a method for high MOI single-cell CRISPR screen analysis. (A complimentary method, scMAGeCK-RRA, is designed for the low-MOI setting.) scMAGeCK-LR (henceforth scMAGeCK) uses a permutation test with ridge regression test statistic to compute $p$-values for pairs of genes and gRNAs. Unfortunately, we were unable to apply scMAGeCK to the real data. First, we were unable to understand the documentation of the scMAGeCK software well enough to confidently apply the method.

Second, scMAGeCK is prohibitively expensive to apply at-scale. The authors of the original scMAGeCK study applied their method only to a small subset of pairs in the Gasperini et al. data. We could not meaningfully compare scMAGeCK to SCEPTRE on calibration and sensitivity metrics without applying scMAGeCK to the full set of gRNA-gene pairs, which, to our knowledge, never has been done (and likely is infeasible).

To enable a simple comparison to scMAGeCK on the simulated data, we implemented a custom, in-house version of scMAGeCK based on a careful examination of the scMAGeCK codebase and a close reading of the original paper. We view this custom implementation as a faithful interpretation of the method in the specialized one-gene to one-NTC setting. We applied our implementation of scMAGeCK to the simulated data, using $B = 1,000$ permutations, the default option. To reduce confusion, we reported the results of the scMAGeCK simulation study in the supplementary materials (Fig. S6) rather than the "Results" section. We could not apply our custom implementation of scMAGeCK to the real data, because the real data are significantly more complex than the simulated data. For example, the real data consist of many genes and gRNAs, and the gRNAs are differently typed (e.g., negative control, positive control, enhancer-targeting, etc.), complicating the analysis considerably.

### Definition of Gasperini et al. discovery set

Gasperini et al. reported a total of 664 gene-enhancer pairs, identifying 470 of these as "high-confidence." We chose to use the latter set, rather than the former, for all our comparisons. Gasperini et al. carried out their ChIP-seq and HI-C enrichment analyses only on the high-confidence discoveries, so for those comparisons we did the same. Furthermore, the 664 total gene-enhancer pairs reported in the original analysis were the result of a BH FDR correction that included not only the candidate enhancers but also hundreds of positive controls. While Bonferroni corrections can only become more conservative when including more hypotheses, BH corrections are known to become anticonservative when extra positive controls are included [27]. To avoid this extra risk of false positives, we chose to use the "high-confidence" set throughout.

### Xie et al. significance scores and discovery set

Xie et al. reported a local (or *cis*) discovery set, which consisted of gene-gRNA pairs with a significance score of greater than zero (see original manuscript for definition of "significance score" [8]; cutoff of zero arbitrary). This discovery set was not directly comparable to the SCEPTRE discovery set. First, the candidate set of *cis* gene-gRNA pairs tested by Xie et al. consisted of gRNAs within *two* Mb of a protein-coding gene *or* long-noncoding RNA. Our candidate *cis* set, by contrast, consisted of gRNAs within *one* Mb of a protein-coding gene. We defined our candidate *cis* set differently than Xie et al. to maintain consistency with Gasperini et al. Second, Xie et al. appear to have used a significantly more conservative threshold than Gasperini et al. in defining their discovery set, but this was challenging to ascertain given the impossibility of FDR correction on the significance scores. To enable a meaningful comparison between Virtual FACS and SCEPTRE, we therefore ranked the Virtual FACS pairs by their significance score and selected the top *n* pairs, where *n* was the size of the SCEPTRE discovery set at FDR 0.1.

### ChIP-seq, HI-C enrichment analyses

ChIP-seq and HI-C enrichment analyses on the Gasperini et al. data (see Figs. 4e, f and S5a) were carried out almost exactly following Gasperini et al. The only change we made is in our quantification of the ChIP-seq enrichment (Fig. 4f). We used the odds ratio of a candidate enhancer being paired to a gene, comparing the top and bottom ChIP-seq quintiles. On the Xie et al. data, we binned the candidate enhancers into two (rather than five) quantiles due to the fewer number of candidate *cis* pairs. We computed odds ratios by comparing enhancers in the upper quantile to those that did not intersect a ChIP-seq peak at all (Fig. S5b).

## Supplementary Information

---

**Additional file 1:** Supplementary figures and tables

**Additional file 2:** Review history

---

### Acknowledgements

We are indebted to Molly Gasperini, Jacob Tome, and Andrew Hill for clarifying several aspects of their data analysis [9] and to the Shendure lab for providing extensive feedback on an earlier draft of this paper. We thank Shiqi Xie for providing guidance on using the Xie et al. 2019 data and Wei Li for a helpful discussion on scMAGeCK. Finally, we thank Tom Norman, Atray Dixit, and Wesley Tansey for useful discussions on single-cell CRISPR screens. This work was supported, in part, by National Institute of Mental Health (NIMH) grant R01MH123184 as well as SFARI Grant 575547. Part of the data analysis used the Extreme Science and Engineering Discovery Environment (XSEDE) [28], which is supported by National Science Foundation grant number ACI-1548562. Specifically, it used the Bridges system [29], which is supported by NSF award number ACI-1445606, at the Pittsburgh Supercomputing Center (PSC).

### Peer review information

### Review history

The review history is available as Additional file 2.

### Authors' information

Twitter handle: @EugeneKatsevich (Eugene Katsevich)

### Authors' contributions

GK and KR identified the problem, conceived the research, and provided supervision. GK developed the method with input from TB and KR. TB and GK implemented the method with assistance from JM. TB, XW, and GK performed the analyses. JM provided guidance on interpretation and communication of results. TB and GK wrote the manuscript with input from all authors. The authors read and approved the final manuscript.

### Availability of data and materials

Analysis results are available online at https://upenn.box.com/v/sceptre-files-v8. All analysis was performed on publicly available data. The Gasperini et al. CRISPR screen data [9] are available at www.ncbi.nlm.nih.gov/geo/query/acc.cgi?acc=GSE120861. The Xie et al. single-cell and bulk CRISPR screen data are available at www.ncbi.nlm.nih.gov/geo/query/acc.cgi?acc=GSE129837. The ChIP-seq data are taken from the ENCODE project [30] and are available at www.encodeproject.org/. The HI-C enrichment analysis is based on the data from Rao et al. [31], available at www.ncbi.nlm.nih.gov/geo/query/acc.cgi?acc=GSE63525. The eQTL and eRNA co-expression *p*-values are taken from the GeneHancer, database [32] available as part of GeneCards (www.genecards.org/).

The `sceptre` R package is available at https://github.com/Katsevich-Lab/sceptre. Vignettes and tutorials are available at https://katsevich-lab.github.io/sceptre/. The scripts used to run the analyses reported in this paper are available at https://github.com/Katsevich-Lab/sceptre-manuscript. (permanent version deposited at Zenodo; DOI 10.5281/zenodo.5643541 [33]). All software is released under an MIT license.

## Declarations

### Ethics approval and consent to participate

Ethical approval not applicable.

### Competing interests

The authors declare that they have no competing financial interests.

**Author details**
[1]Department of Statistics and Data Science, Carnegie Mellon University, Pittsburgh, PA 15213, USA. [2]New York Genome Center, New York, USA. [3]Department of Biology, New York University, 24 Waverly Pl 6th Floor, New York, NY 10003, USA. [4]Computational Biology Department, Carnegie Mellon University, Pittsburgh, PA 15213, USA. [5]Department of Statistics and Data Science, Wharton School, University of Pennsylvania, Philadelphia, PA 19104, USA.

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

## 

