## [**Additional file 2** Review history · Genome Biology]

Review History

First round of review

Reviewer 1

Were you able to assess all statistics in the manuscript, including the appropriateness of statistical tests used? No.

Were you able to directly test the methods? No.

Comments to author:

The manuscript entitled "Conditional resampling improves calibration and sensitivity in single-cell CRISPR screen analysis" proposed an analysis pipeline for dissecting association between gRNA perturbation and gene expression from single cell CRISPR screen data. The authors employed conditional resampling approach to improve the statistics power of the analysis by adjusting the technical confounder and facilitating calibration test of association. This work focuses on an important issue in single cell CRISPR screens. However, additional work should be done to fully validate the improvement of this method over the existing approaches.

1. The power of the proposed method should be examined using more real datasets, in addition to simulated data and two public studies interrogated in this study. This approach should be not only applicable to enhancer-gene association but also extended to any perturbation-expression association studies. Compared to enhancer-gene pair that lacks enough validated positive controls, CRISPR-based gene perturbation single cell screens may provide more robust true positive/negative hits to evaluate the power of the analytic approaches. It may be worth checking in these datasets and comparing to other statistic models.
2. Fig. 3a-c, the color legend should be indicated for 3a and 3b, but not just confined to 3c.
3. In Fig. 3b-c, scMAGeCK-LR should be also tested with real data despite its poor performance with simulated data.
4. Fig. 3e, in addition to this specific positive control hit, how is the general landscape for the positive control pairs using SCEPERE with Xie et al dataset.
5. Fig. 4 and 5, it is insufficient to draw a conclusion by just comparing SCEPERE with original analytic methods in the two public studies. For each dataset, if applying SCEPERE, monocle NB, improved NB, virtual FACS and scMAGeCK-LR methods, would SCEPERE be still the superior or significantly better than others? Similar comparison could be made using more datasets performing either enhancer or gene perturbation single cell CRISPR screens.
6. If binning the genes by the expression level, how is performance of SCEPERE to pinpoint perturbation-gene pair?
7. Line 253-258, Figure 4 is uncorrected cited which should be Figure 5.

8. Figure S2 and S4 are not cited in the main text.
9. Figure S3, Is there similar analysis for Xie et al data?

Reviewer 2

Were you able to assess all statistics in the manuscript, including the appropriateness of statistical tests used? No.

Were you able to directly test the methods? No.

Comments to author:

Barry and his colleagues conducted a method to improve the calibration and sensitivity in single-cell CRISPR screen analysis. They showed the analysis challenges and demonstrated how they address them by conditional resampling. This work would be interesting and important for the future single cell CRISPR screening. However, some specific points below should be addressed before the publication.

1. Could the author use a summary diagram to demonstrate the analysis challenge and how their method address it?
2. The author should be careful about some terminologies. HiC and ChIP-seq are not functional assays.
3. The author should describe more details in the figure legends so that they will be more readable.
4. In the discussion part, the authors applied SCERTRE to interpret GWAS variant. This is an important part to show

Statement of Revision

In the following, we provide a detailed account of the changes that we have made in the revised paper as well as responses to all comments. We have structured this list into blocks, corresponding to the comments made by the Reviewers. The comments are in blue and our responses are in black.

Comments by Reviewer 1

[R1: 1] “THE POWER OF THE PROPOSED METHOD SHOULD BE EXAMINED USING MORE REAL DATASETS, IN ADDITION TO SIMULATED DATA AND TWO PUBLIC STUDIES INTERROGATED IN THIS STUDY. THIS APPROACH SHOULD BE NOT ONLY APPLICABLE TO ENHANCER-GENE ASSOCIATION BUT ALSO EXTENDED TO ANY PERTURBATION-EXPRESSION ASSOCIATION STUDIES. COMPARED TO ENHANCER-GENE PAIR THAT LACKS ENOUGH VALIDATED POSITIVE CONTROLS, CRISPR-BASED GENE PERTURBATION SINGLE CELL SCREENS MAY PROVIDE MORE ROBUST TRUE POSITIVE/NEGATIVE HITS TO EVALUATE THE POWER OF THE ANALYTIC APPROACHES. IT MAY BE WORTH CHECKING IN THESE DATASETS AND COMPARING TO OTHER STATISTIC MODELS.”

Thank you for this comment. We have applied SCEPTRE to one simulated dataset and three (not two) real, large-scale single-cell CRISPR screen datasets (namely, Gasperini et al. (2019); Xie et al. (2019); Morris et al. (2021)). The real datasets that we examined targeted both enhancers and gene TSSs. For example, Gasperini et al. targeted 381 gene TSSs (corresponding to approximately 2% of all protein-coding genes) and ten previously-validated enhancers, and Morris et al. targeted seven gene TSSs. SCEPTRE was highly sensitive to these positive control perturbations (Figure 3d; Figure S3 in Morris et al.) while retaining type-I error control on the negative control perturbations (Figures 3b-c; Figure 2c in Morris et al.). The real datasets that we examined were generated using different experimental protocols (perturb-seq and ECCITE-seq), demonstrating versatility of the proposed method. We described the application of SCEPTRE to the Morris et al. data in detail in the Discussion section; we did not include displays presenting those results, however, to avoid overlap with the Morris et al. study.

With one exception (Schraivogel et al., 2020), enhancer-targeting single-cell CRISPR screens have been conducted at high-MOI (Gasperini et al. (2019); Xie et al. (2019); Morris et al. (2021)). The statistical challenges at play in the low-MOI setting, while related to those in high MOI setting, are sufficiently distinct that it is most appropriate to defer rigorous analysis of the low MOI setting to a future work. We currently are working on this extension for a followup study that we expect to complete in 6-12 months.

[R1: 2] “FIG. 3A-C, THE COLOR LEGEND SHOULD BE INDICATED FOR 3A AND 3B, BUT NOT JUST CONFINED TO 3C.”

Thank you for this suggestion; we have implemented it in the revised manuscript.

[R1: 3] “IN FIG. 3B-C, scMAGeCK-LR SHOULD BE ALSO TESTED WITH REAL DATA DESPITE ITS POOR PERFORMANCE WITH SIMULATED DATA.”

We were unable to apply scMAGeCK to the real data. First, we found the documentation of the scMAGeCK software package unclear to the point that we were uncertain how to apply the method to new data, despite having examined closely the examples and source code. Second, the authors of the original study deployed scMAGeCK only to a small subset of the gene-gRNA pairs assayed in Gasperini et al. due to the prohibitive computational burden of the method. To the best of our knowledge, scMAGeCK has never been deployed at-scale to the Gasperini et al. data, which would be necessary to meaningfully compare SCEPTRE to scMAGeCK on calibration and sensitivity metrics.

The previous version of the manuscript reported the output of scMAGeCK on a simple, simulated dataset consisting of one gene and one negative control gRNA (Figure 3a). We did not apply scMAGeCK “out

of the box” to this simulated dataset, because of the limitations of the documentation. Rather, we implemented a custom, in-house version of scMAGeCK that consisted of code snippets from the scMAGeCK codebase. Our custom implementation is a faithful interpretation of the method in the specialized one-gene to one-NTC setting. However, our custom implementation does not apply to real data, because the real data are considerably more complex than the simulated data. For example, the real data consist of many genes and gRNAs, and the gRNAs come in different types (e.g., negative control, positive control, enhancer-targeting, etc.), complicating the analysis considerably.

In sum, we cannot apply scMAGeCK to the real data due to documentation and computational challenges. To reduce confusion, we described our challenges with scMAGeCK in the Methods section, and we moved the simulation result to the supplementary materials.

[R1:4] “FIG. 3E, IN ADDITION TO THIS SPECIFIC POSITIVE CONTROL HIT, HOW IS THE GENERAL LANDSCAPE FOR THE POSITIVE CONTROL PAIRS USING SCEPERE WITH XIE ET AL DATASET.”

Unlike the Gasperini data, the Xie data did not come with any explicit positive control perturbations. In lieu of positive controls, Xie et al. (2019) conducted singleton perturbation screens using a highly sensitive bulk RNA-seq assay. These bulk screens were conducted only for perturbations targeting ARL15-enh and MYB-enh-3. We excluded the latter perturbations from consideration because these were found by Xie et al. to have fitness effects (see Figure S1A of Xie et al. (2019)), a complication beyond the scope of the current work. In short, Figure 3e represents the only positive control validation we could perform on the Xie dataset.

[R1:5] “FIG. 4 AND 5, IT IS INSUFFICIENT TO DRAW A CONCLUSION BY JUST COMPARING SCEPERE WITH ORIGINAL ANALYTIC METHODS IN THE TWO PUBLIC STUDIES. FOR EACH DATASET, IF APPLYING SCEPERE, MONOCLE NB, IMPROVED NB, VIRTUAL FACS AND scMAGeCK-LR METHODS, WOULD SCEPERE BE STILL THE SUPERIOR OR SIGNIFICANTLY BETTER THAN OTHERS? SIMILAR COMPARISON COULD BE MADE USING MORE DATASETS PERFORMING EITHER ENHANCER OR GENE PERTURBATION SINGLE CELL CRISPR SCREENS.”

We conducted a new analysis in which we applied Monocle NB to the Xie data as well as improved NB to both datasets; Figures 4 and 5 in the revision reflect these changes. Unfortunately, Virtual FACS is not implemented as a publicly available software, so we could not apply it to the Gasperini data. We found that SCEPTRE remained the best method despite the addition of these competitor methods. We updated the text in the section “Analysis of candidate cis-regulatory pairs” to reflect these new analyses.

[R1:6] “IF BINNING THE GENES BY THE EXPRESSION LEVEL, HOW IS PERFORMANCE OF SCEPERE TO PINPOINT PERTURBATION-GENE PAIR?”

We conducted a new analysis in which we binned candidate gene-enhancer pairs by gene expression level and computed the fraction of pairs rejected in each bin (Tables S1 and S2). We found that SCEPTRE was more likely to reject pairs that contained more highly-expressed genes. We replicated this analysis on other methods (not shown) and observed similar trends. We added a paragraph to the section “Analysis of candidate cis-regulatory pairs” summarizing these results.

[R1:7] “LINE 253-258, FIGURE 4 IS UNCORRECTED CITED WHICH SHOULD BE FIGURE 5.”

Thank you for pointing this out; we have now fixed this error.

[R1:8] “FIGURE S2 AND S4 ARE NOT CITED IN THE MAIN TEXT.”

We now cite Figure S2 in lines 161-163:

“Finally, we compute a left-, right-, or two-tailed probability of the original z-value under the empirical null distribution, yielding a well-calibrated p-value. This p-value can deviate substantially from that obtained based on the standard normal (Figure 2, Figure S2).”

We now cite Figure S5 (formerly Figure S4) in line 242 and in line 269:

“Finally, enhancers discovered by SCEPTRE showed improved enrichment across all eight cell-type relevant ChIP-seq targets reported by Gasperini et al. (Figure 4d, Figure S5a).”

“The SCEPTRE discoveries were more biologically plausible: compared to the Virtual FACS pairs, the SCEPTRE pairs were (i) physically closer (Figure 5b), (ii) more likely to fall within the same TAD (Figure 5c), (iii) more likely to interact when in the same TAD (Figure 5c), and (iv) more enriched for all eight cell-type relevant ChIP-seq targets (Figure 5d and Figure S5b).”

[R1: 9] “FIGURE S3, IS THERE SIMILAR ANALYSIS FOR XIE ET AL DATA?”

We have added Figure S4, which is the analog of Figure S3 for the Xie data.

Comments by Reviewer 2

[R1: 1] “COULD THE AUTHOR USE A SUMMARY DIAGRAM TO DEMONSTRATE THE ANALYSIS CHALLENGE AND HOW THEIR METHOD ADDRESS IT?”

We added a table (Table 1) in the “Analysis Challenges” section summarizing the analysis challenges and how the proposed method addresses them. The table conveys that (i) parametric methods struggle with gene expression distribution misspecification, (ii) nonparametric methods struggle with confounder adjustment, and (iii) conditional resampling addresses both of these challenges.

[R1: 2] “THE AUTHOR SHOULD BE CAREFUL ABOUT SOME TERMINOLOGIES. HiC AND CHIP-SEQ ARE NOT FUNCTIONAL ASSAYS.”

We have replaced the terms “functional evidence” and “functional assays” with “biological evidence” and “orthogonal assays” throughout the manuscript. This clarifies that HiC and ChIP-seq are not functional assays.

[R1: 3] “THE AUTHOR SHOULD DESCRIBE MORE DETAILS IN THE FIGURE LEGENDS SO THAT THEY WILL BE MORE READABLE.”

We expanded and clarified several of the figure legends. For example, in the legend of Figure 3, we clarify that the negative binomial model is correctly specified on three of the four simulated datasets, and that only SCEPTRE maintained calibration across all four simulated datasets despite model misspecification and confounder presence. Additionally, in the legend of Figure 5, we clarify that the SCEPTRE and Virtual FACS discovery sets diverged substantially, and that SCEPTRE pairs showed enrichment for biological signals associated with enhancer activity on (b) physical distance, (b) HiC interaction, and (d) ChIP-seq enrichment metrics.

[R1: 4] “IN THE DISCUSSION PART, THE AUTHORS APPLIED SCEPTRE TO INTERPRET GWAS VARIANT. THIS IS AN IMPORTANT PART TO SHOW”

We agree that it is important to highlight the application of SCEPTRE to the new STING-seq data for interpreting GWAS variants—which we do in both the introduction and the discussion. We do not include any figures presenting these results, however, to avoid overlap with the STING-seq preprint (Morris et al., 2021).

References

Gasperini, M., Hill, A. J., McFaline-Figueroa, J. L., Martin, B., Kim, S., Zhang, M. D., Jackson, D., Leith, A., Schreiber, J., Noble, W. S., Trapnell, C., Ahituv, N., and Shendure, J. (2019). A Genome-wide Framework for Mapping Gene Regulation via Cellular Genetic Screens. *Cell*, 176(1-2):377–390.e19.

- Morris, J. A., Daniloski, Z., Domingo, J., Barry, T., Ziosi, M., Glinos, D. A., Hao, S., Mimitou, E., Smibert, P., Roeder, K., et al. (2021). Discovery of target genes and pathways of blood trait loci using pooled crispr screens and single cell rna sequencing. bioRxiv.
- Schraivogel, D., Gschwind, A. R., Milbank, J. H., Leonce, D. R., Jakob, P., Mathur, L., Korbel, J. O., Merten, C. A., Velten, L., and Steinmetz, L. M. (2020). Targeted Perturb-seq enables genome-scale genetic screens in single cells. *Nature Methods*, 17(6):629–635.
- Xie, S., Armendariz, D., Zhou, P., Duan, J., and Hon, G. C. (2019). Global Analysis of Enhancer Targets Reveals Convergent Enhancer-Driven Regulatory Modules. *Cell Reports*, 29(9):2570–2578.e5.

Second round of review

Reviewer 1

Concerns have been satisfactorily addressed.

Reviewer 2

I still think that interpreting GWAS variants is an important part to show the performance and the application of the tool. Authors should include it in this manuscript.